# Red Blood Cell Lifespan < 74 Days Can Clinically Reduce Hb1Ac Levels in Type 2 Diabetes

**DOI:** 10.3390/jpm12101738

**Published:** 2022-10-19

**Authors:** Saijun Zhou, Rongna Dong, Junmei Wang, Li Zhang, Bai Yu, Xian Shao, Pufei Bai, Rui Zhang, Yongjian Ma, Pei Yu

**Affiliations:** 1NHC Key Laboratory of Hormones and Development, Chu Hsien-I Memorial Hospital and Tianjin Institute of Endocrinology, Tianjin Medical University, Tianjin 300134, China; 2Tianjin Key Laboratory of Metabolic Diseases, Tianjin Medical University, Tianjin 300134, China; 3Guangdong Breath Test Engineering and Technology Research Center, Shenzhen University, Shenzhen 518000, China

**Keywords:** average blood glucose, glycosylated hemoglobin, red blood cell lifespan, type 2 diabetes

## Abstract

Variations in the red blood cell (RBC) lifespan can affect glycosylated hemoglobin (HbA1c) test values, but there is still a lack of evidence regarding how and to what degree the RBC lifespan influences HbA1c in the type 2 diabetes mellitus (T2DM) population owing to the restriction of traditional RBC lifespan detection means. In this study, we monitored 464 T2DM patients and 231 healthy control finger blood glucose levels at seven time points for three consecutive months. The HbA1c levels were assessed at the end of the third month as well as the RBC lifespan was measured through the CO breath test. T2DM patients were stratified into four quartile groups according to their RBC lifespans. There was no statistical significance in HbA1c among these four groups. However, the average blood glucose in the Q1 group was significantly higher than those in the other groups. Additionally, the contribution of RBC lifespan to HbA1c test value in the Q1 group was 14.07%, which was significantly higher than those in the other groups. Finally, we used multiple linear regression models to construct a mathematical formula to correct the HbA1c test value in the Q1 group, which would benefit the management of T2DM.

## 1. Introduction

Type 2 diabetes mellitus (T2DM) has become one of the chronic diseases that seriously threaten people’s health worldwide. A recent epidemiological survey showed that the number of patients with T2DM worldwide has exceeded 460 million, and the prevalence continues to increase year by year [1]; so, the clinical prevention and treatment of T2DM are arduous tasks. The glycosylated hemoglobin (HbA1c) value can reflect the blood glucose control status of patients over the past 2–3 months, which is not only an important indicator in the diagnosis of T2DM but is also regarded as both the gold standard for evaluating whether the blood glucose management of diabetic patients achieves the target value and an important basis for guiding clinical adjustments to hypoglycemic therapy [2]. A large number of evidence-based medical studies have confirmed that the attainment of HbA1c in patients with T2DM can significantly improve the risk profile and long-term prognosis of chronic complications among diabetic patients [3,4,5]. Vascular complications of both the macrovascular system (cardiovascular disease (CVD)) [6,7] and microvascular system (diabetic kidney disease (DKD) [8], diabetic retinopathy [9] and neuropathy [10]) are the leading cause of morbidity and mortality in individuals with diabetes. Therefore, whether the detection value of HbA1c can accurately reflect the average blood glucose level of patients is of great significance for the clinical management of T2DM [11].

Accumulating evidence suggests that the level of HbA1c not only depends on the average blood glucose level but is also affected by the lifespan of red blood cells (RBCs) [1,2]. Malka et al. indicated that the differences in RBC lifespan can explain almost 100% of the variation in HbA1c, independent of the blood glucose level [12]. However, there is still a lack of evidence about how and to what degree the RBC lifespan affects the HbA1c value in the T2DM population. In this study, a CO breath test, which is a reliable method for quickly determining the human RBC lifespan for clinical applications [13,14,15,16], was used to measure the RBC lifespan of patients with T2DM, and the relationship between the HbA1c test value and the average blood glucose level of patients with T2DM in different RBC lifespan groups was analyzed to provide useful guidance for accurate clinical evaluation of blood glucose control in patients with T2DM.

## 2. Methods

### 2.1. Participants

We included 464 patients with T2DM who were admitted to the “Trinity Care” outpatient clinic and 231 healthy controls from healthy care center of Chu Hsien-I Memorial Hospital, Tianjin Medical University, from September 2019 to January 2021. We monitored each patient’s finger blood glucose value at seven time points using our hospital’s blood glucose management system for three consecutive months. At the end of the third month, we collected their blood to test HbA1c and Hb levels, and we used alveolar gas samples to measure their RBC lifespan.

Inclusion criteria were (1) adult patients with T2DM older than 18 years of age, diagnosed according to the 2015 American Diabetes Association diagnostic criteria; (2) received T2DM diagnosis and at least 3 months of treatment at our hospital; (3) had been using a blood glucose monitoring system for the past 3 months; (4) underwent monitoring of finger blood glucose before and 2 h after each meal, and before sleeping at least twice a week, according to the A1C-derived Average Glucose (ADAG) study [17]; (3) blood glucose levels had not fluctuated considerably in the past 3 months, and the difference between maximum and minimum blood glucose levels was less than 4.4 mmol/L; (4) voluntary participation in and cooperation with the study and signed informed consent. Exclusion criteria were (1) type 1 diabetes; (2) high blood glucose fluctuation (the difference between the maximum and minimum blood sugar value within a day is greater than 4.4 mmol/L), repeated hypoglycemic attacks (once a week on average); (3) pregnant or breastfeeding women; severe diabetic acute complications such as diabetic ketoacidosis, hyperosmotic hyperglycemia syndrome, or lactic acidosis; (4) hemolytic anemia leukemia; history of blood donation and transfusion in the past 4 months; (5) malignant neoplasms, active infection; hyperthyroidism; (6) chronic liver diseases or alanine aminotransferase (ALT) increased by more than three times the normal levels; (7) chronic kidney disease (creatinine > 1.5 mg/dL); (8) after heart valve replacement or New York Heart Association functional grade III or above; (9) smoking; (10) impaired lung function; (11) use of drugs that may affect the RBC lifespan, such as ribavirin [18], barbiturates, or phenobarbital sodium [19]. This study was approved by the ethics committee of Chu Hsien-I Memorial Hospital of Tianjin Medical University. All patients provided their written informed consent.

### 2.2. Data Collection and Laboratory Examination

Using medical records, we collected clinical data, including age, sex, duration of diabetes, and medications, and blood glucose values using the iHealth software of the “Trinity Care” patient management system of our hospital, and weight, height, and brachial artery blood pressure of the right upper limb were measured by trained personnel. Fasting venous blood was collected and tested at the central laboratory of Chu Hsien-I Memorial Hospital, Tianjin Medical University. A Beckman Coulter AU5800 automatic biochemical analyzer (Beckman Coulter, Kraemer Blvd, Brea, CA, USA) was used to detect glycosylated serum albumin (GA), liver and kidney function, and blood lipids. Patients’ HbA1c levels were tested using a Sysmex XN instrument (Sysmex, Kobe, Japan). HbA1c test program conformed to the standard of the National Glycohemoglobin Standardization Program.

### 2.3. Mean Blood Glucose

All eligible patients with T2DM who performed self-blood glucose monitoring using the “Trinity Care” outpatient and diabetes management system for 3 consecutive months were included in this study. The frequency of blood glucose monitoring followed that of the ADAG study [17], and each patient monitored their finger blood glucose value at seven time points (before and after each meal plus bedtime) level at least twice every week, and the number of blood glucose values was ≥14 per week. The total number of blood glucose values per patient included in the final analysis was ≥168. The average blood glucose (AG) calculation method was according to the ADAG study [17]—that is, the arithmetic average of blood glucose measurement values for each patient.

### 2.4. The CO Breath Test to Measure the RBC Lifespan

Levitt’s CO breath test was carried out to detect the RBC lifespan. The detection protocol was designed according to studies recently published [13,16]. In brief, we collected each participant’s alveolar gas samples at 8:00–10:00 a.m. after an overnight fast and a 20 min rest period. Each participant was instructed to take a deep breath, hold it for 15 s, and then exhale into an alveolar gas–collection system through the mouthpiece (Figure 1). Alveolar gas and atmospheric gas samples were stored at room temperature and analyzed within 24 h. The ELS Tester (Seekya Biotec Co., Ltd., Shenzhen, China), an automated instrument, was used to determine CO concentrations, and it was used to calculate the RBC lifespan with Levitt’s formula [13,16].

### 2.5. Statistical Analysis

Statistical analysis was conducted using IBM SPSS version 25.0 software program (IBM Corp., Armonk, NY, USA) and R version 4.0.3 software program (The R Project for Statistical Computing, Vienna, Austria). A *t*-test was used to compare data groups with a normal distribution, while a chi-squared test was used to compare data groups with a non-normal distribution. Pearson’s correlation analysis and restrictive spline regression were used to fit the relationship between continuous variables. Two-sided *p* values < 0.05 were considered statistically significant. Multiple linear regression was used to assess both AG and the RBC lifespan’s contributions to HbA1c value and to establish a mathematical formula to correct the effect of a shorter RBC lifespan on HbA1c detection values.

## 3. Results

### 3.1. Clinical Data of Patients with T2DM

The general clinical characteristics of 464 patients with T2DM and 231 healthy controls, with an average age of 56.8 ± 11.8 and 55.0 ± 9.9 years, respectively. As shown in Table 1, the median duration of T2DM was 6 years (IQR, 2.9–12 years). The mean fasting blood glucose (FBG) values in diabetic patients and the healthy control group were 7.57 ± 1.70 mmol/L and 5.06 ± 0.39 mmol/L, and the Hb concentrations were 142.8 ± 18.8 g/L and 139.1 ± 17.3 g/L, respectively. There were no significant statistical differences with regard to age, sex, and liver and kidney functions, while the FBG level, systolic blood pressure (SBP), diastolic blood pressure (DBP), blood lipid levels in patients with T2DM were significantly higher than those in the healthy control group (all *p* < 0.05).

We stratified patients with T2DM into the following four quartile groups based on their RBC lifespan: Q1, RBC lifespan of fewer than 75 days; Q2, RBC lifespan of 75–89 days; Q3, RBC lifespan of 90–111 days; and Q4, RBC lifespan of at least 112 days. There were no significant statistical differences with regard to the duration of diabetes, SBP, DBP, Hb, blood lipids, and liver and kidney functions among the four groups. The FBG and 2 h postprandial blood glucose (P2BG) levels in the Q1 group were significantly higher than those in the Q2, Q3, and Q4 groups (all *p* < 0.05), and those in the Q2 and Q3 groups were significantly higher than that in the Q4 group (all *p* < 0.05). The percentage of men in the Q1 group was significantly higher than those in the Q2, Q3, and Q4 groups (all *p* < 0.05).

### 3.2. The RBC Lifespan Distribution in Patients with T2DM

The mean RBC lifespan in patients with T2DM (95.0 ± 30.3 days; 95% confidence interval [CI], 35.6–154.4 days) was significantly lower than that in the healthy control group (100.1 ± 30.9 days; 95% CI, 39.6–160.1 days) (*p* = 0.0433) (Figure 2A). Additionally, as shown in Figure 2B, the mean RBC lifespan of men (90.4 ± 30.4 days) was significantly lower than that of women (99.7 ± 29.5 days) in patients with T2DM (*p* = 0.0009), and there was also a tendency toward a reduction in the healthy control group.

### 3.3. Correlation between the RBC Lifespan and Clinical Characteristics of T2DM

As shown above, the RBC lifespan was significantly shorter in patients with T2DM. We further analyzed the relationship between the RBC lifespan and clinical indices of T2DM. Pearson’s analysis showed that there was a significant negative correlation between the RBC lifespan and AG level (r = −0.33; *p* < 0.0001) and glycosylated albumin (GA) and (r = −0.14; *p* = 0.010), respectively, while there was no significant correlation with age, HbA1c, duration of diabetes, or Hb (all *p* > 0.05), as shown in Table 2.

### 3.4. The Effect of Different RBC Lifespans on HbA1c Test Value

To analyze how the RBC lifespan affects the HbA1c test value in patients with T2DM, the AG level of patients in the four groups was compared firstly. As shown in Figure 3A, across the quartile groups (in order from Q1 to Q4), the AG level of patients decreased successively, and the differences were statistically significant (Q1 vs. Q2, *p* = 0.031; Q1 vs. Q4, *p* < 0.0001; Q2 vs. Q4, *p* < 0.1; Q3 vs. Q4, *p* = 0.0004). There was no significant difference between HbA1c values among the four groups (*p* > 0.05).

### 3.5. The Contribution of the RBC Lifespan to HbA1c Test Value

To further assess what degree the RBC lifespan affects the HbA1c test value, we further assessed both AG and the RBC lifespan’s contributions to HbA1c value. As shown in Figure 4, the contribution of the RBC lifespan of all patients with T2DM was 6.5%, while that in the Q1 group was 14.7% and that in the Q2-Q4 groups were about only 1%. These results indicate that only shorter RBC lifespans could cause a clinically significant HbA1c difference.

### 3.6. Adjustment of the Effect of the RBC Lifespan on HbA1c Detection Values

As shown above, the RBC lifespan in the Q1 group could have a clinically significant influence on HbA1c detection values, so it is very necessary to correct this influence. We used a multivariate linear regression model to establish a mathematical formula to correct the effect of a shorter RBC lifespan on HbA1c detection values. In the group with an RBC lifespan less than 75 days, as shown in Figure 5, HbA1c_(corrected)_ = −0.035367 × RBCs + 1.010689 × HbA1c_(t)_ + 2.953272. The R² of the correction formula is 0.772, and the adjusted R² = 0.766.

## 4. Discussion

Important clinical decisions for millions of patients with diabetes rely on measurements of blood HbA1c. However, the RBC lifespan could have a clinically significant influence on HbA1c detection values, so it is very necessary to correct this influence. Accumulating evidence suggests that the level of HbA1c depends not only on the average blood glucose level but also on the lifespan of RBCs [1,2]. Malka et al. indicated that interpatient variation in derived mean red blood cell age can explain all glucose-independent variation in HbA1c [12], and Kameyama et al. showed that the differences in HbA1c for a given mean blood glucose value can be explained by the average age of RBC lifespan [20]. Khera PK et al. investigated the relationship between RBC lifespan and HbA1c in a group of nine hematologically normal diabetic and nondiabetic subjects using oral 15 N-glycine to label heme in an age cohort of RBC [21]. The limitations are that these research samples were too small and did not reveal how to adjust the influence of RBC lifespan on HbA1c. Future testing in larger populations is thus necessary. Chu HW et al. studied the relationship between single fasting blood glucose, HbA1c and RBC lifespan in a large sample size [16]. However, the limitation is that fasting blood glucose has diurnal variability, which cannot accurately reflect the average blood glucose level of diabetic patients for 3 months. So there is still a lack of studies on how and to what degree the influence of the RBC lifespan variation on HbA1c detection value in a large sample of the T2DM population. In this study, a unified standard blood glucose monitoring system was adopted to completely collect the blood glucose monitoring values of T2DM patients within 3 months, which can better indicate the average blood glucose level of patients, so as to more accurately reflect the influence of RBC lifespan on HbA1c. We clarified the extent of the RBC lifespan of patients with T2DM affected the HbA1c detection value and attempted to adjust this impact.

We found that the mean RBC lifespan of men (90.4 ± 30.4 days) was significantly lower than that of women (99.7 ± 29.5 days) in patients with T2DM, which may be the cause of the difference in Hb concentration between genders and also gender difference in T2DM complications [11,22]. Patients with RBC lifespan ≤ 74 days (in the Q1 group), the contribution of RBC lifespan was 14.7%, while that in the Q2–Q4 groups were about only 1%. These results indicate that only a shorter RBC lifespan could cause a clinically significant difference in HbA1c, which can be explained by the fact of insufficient glycosylation of Hb exposed to certain blood glucose levels. This phenomenon can be explained by the fact that the shorter RBC lifespan results in insufficient glycosylation of Hb exposed to certain blood glucose levels. Tahara et al. elucidated the relationship between the change in HbA1c and the time after exposure to a certain blood glucose level, and they also found that the average time it takes for the HbA1c values of T2DM patients with baseline HbA1c concentrations of more than 12% to drop by 90% is 70.9 ± 12.9 days [23]. This study suggested that the time for Hb to reach a sufficient glycosylation level is about 70 days at a certain blood glucose level. Additionally, this result was consistent with our study, both of which strongly indicated the HbA1c test values in T2DM with a RBC lifespan less than 74 days cannot truly reflect their blood glucose levels, resulting in long-term poor blood glucose control. Therefore, it is of great importance to get true HbA1c values for reducing diabetic vascular complications and improving the prognosis of them. So We used a multivariate linear regression model to establish a mathematical formula to correct the effect of a shorter RBC lifespan on HbA1c detection values. In the group with an RBC lifespan ≤ 74 days, we can use the formula: HbA1c (corrected) = −0.035367 × RBCs + 1.010689 × HbA1c (test) + 2.953272. The R² of the correction formula is 0.772, and the adjusted R² = 0.766. Therefore, this study strongly indicated that **i**n clinical practice, to accurately assess blood glucose levels in patients with T2DM, we need both the traditional HbA1c and RBC lifespan values. When a patient’s RBC lifespan is less than 75 days, we need to calculate their true HbA1c by using the above formula, and the above mentioned formula will provide useful clinical information to evaluate patients’ blood glucose levels more accurately.

The main finding of this study is that only a shorter RBC lifespan could trigger clinically significant changes in HbA1c value. A shorter RBC lifespan can result in HbA1c test values that significantly underestimate the true blood glucose level of patients with T2DM. The limitations of this study include a relatively modest sample size, and this was a single-center study. The correction formula must be further verified in a larger, multi-center study.

In conclusion, shorter RBC lifespan could have a clinically significant influence on glycosylated hemoglobin detection values, so it is very necessary to correct this influence. Measuring the influence of the RBC lifespan on HbA1c will benefit the management of T2DM and provide useful guidance for the accurate clinical evaluation of blood glucose control in patients with T2DM.

## Figures and Tables

**Figure 1 jpm-12-01738-f001:**
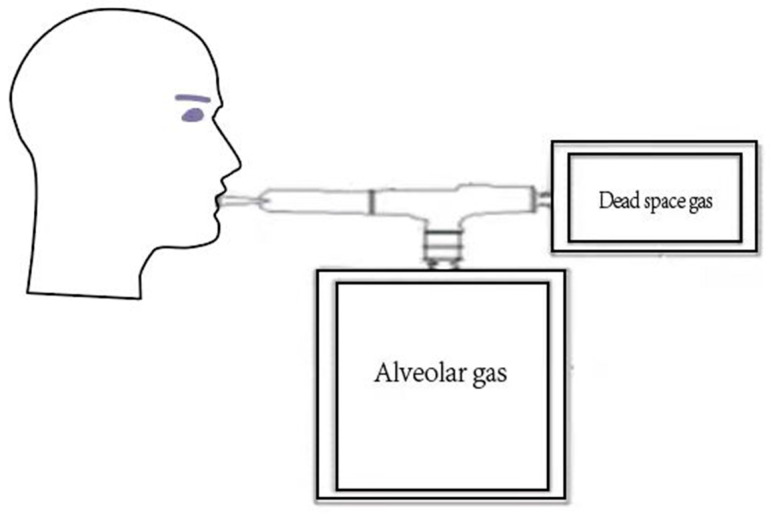
Alveolar gas collecting system.

**Figure 2 jpm-12-01738-f002:**
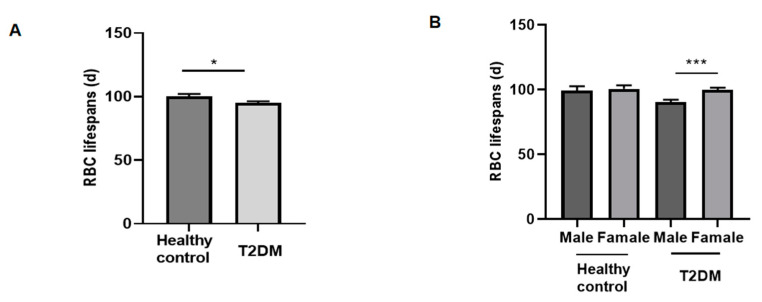
Comparison of RBC lifespans between healthy control and patients with type 2 diabetes mellitus (T2DM): (**A**). Comparison of red blood cell (RBC) lifespans in the two groups. (**B**). Comparison of red blood cell (RBC) lifespans in male and female of two groups, respectively.* *p* < 0.1, *** *p* < 0.001.

**Figure 3 jpm-12-01738-f003:**
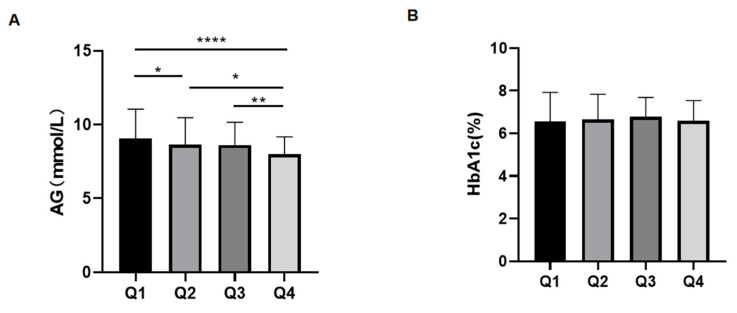
Comparison of average glucose (AG) and glycosylated hemoglobin (HbA1c) among the four groups of patients with type 2 diabetes mellitus (T2DM): (**A**). Comparison of AG among the four groups. (**B**). Comparison of HbA1c among the four groups. * *p* < 0.1 ** *p* < 0.01 **** *p* < 0.0001.

**Figure 4 jpm-12-01738-f004:**
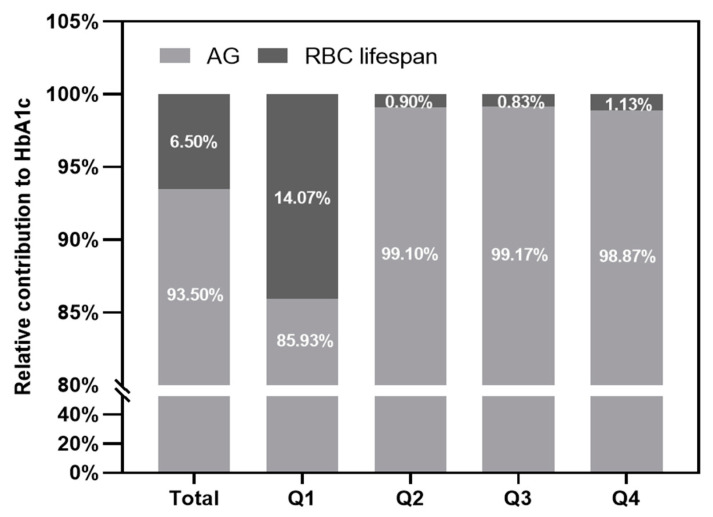
The contributions of average glucose (AG) and red blood cell (RBC) lifespan to glycosylated hemoglobin (HbA1c) test value, respectively.

**Figure 5 jpm-12-01738-f005:**
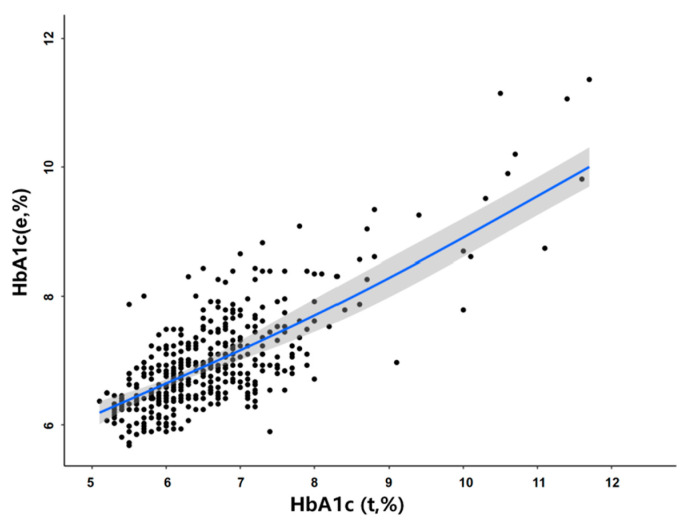
Multivariate linear regression relationship between glycosylated hemoglobin detection value and HbA1c estimated value in patients with type 2 diabetes mellitus (T2DM). HbA1c (estimated)% = (average blood glucose + 2.5944)/1.5944). HbA1c (corrected) = −0.035367 × RBCs + 1.010689 × HbA1c (test) + 2.953272 (R^2^ = 0.772).

**Table 1 jpm-12-01738-t001:** The clinical characteristics of healthy control and T2DM patients.

Parameters	Healthy Control	T2DM Total	T2DM RBC Lifespan Quartile				^a^ *p*	^b^ *p*
			Q1: <75 d	Q2: 75–89d	Q3: 90–111d	Q4: ≥112d		
N	231	464	113	113	120	118	0.0553	0.520
Sex (M/F)	(100/115)	(233/231)	(71/42)	(57/56)	(58/62)	(46/72)	0.369	0.006
Duration (year)	0	6 (2.9, 12)	6.5 (3, 17.4)	7 (2, 13)	6 (2.8, 12)	6.8 (2, 12)	-	0.240
SBP (mmHg)	128.1 ± 12.9	135.5 ± 16.0	139.2 ± 18.1	134.2 ± 13.5	134.4 ± 14.7	134.1 ± 16.7	<0.0001	0.05
DBP (mmHg)	77.5 ± 10.3	80.4 ± 11.2	82.9 ± 12.5	78.7 ± 10.1	79.4 ± 10.6	80.5 ± 11.2	0.0012	0.05
Hb (g/L)	139.1 ± 17.3	142.8 ± 18.8	140.7 ± 21.7	143.1 ± 18.9	144.7 ± 16.0	143.1 ± 17.8	0.0014	0.467
FBG (mmol/L)	5.06 ± 0.39	7.57 ± 1.70	8.27 ± 2.47	7.55 ± 1.51	7.49 ± 1.20	7.06 ± 1.29	<0.0001	<0.0001
P2BG (mmol/L)	-	9.75 ± 2.17	10.62 ± 2.46	9.87 ± 2.38	9.70 ± 1.86	8.94 ± 1.65	-	<0.0001
TG (mmol/L)	1.24 ± 0.50	1.82 ± 1.16	1.94 ± 1.20	1.76 ± 1.06	1.90 ± 1.42	1.67 ± 0.87	<0.0001	0.2469
TC (mmol/L)	4.70 ± 0.91	4.91 ± 1.11	5.19 ± 1.14	4.71 ± 1.21	4.94 ± 1.05	4.77 ± 0.99	0.0157	0.0596
LDL-c (mmol/L)	2.93 ± 0.68	3.25 ± 0.89	3.48 ± 0.92	3.06 ± 0.99	3.25 ± 0.82	3.19 ± 0.79	<0.0001	0.0503
ALT (U/L)	24.2 ± 12.7	24.5 ± 15.5	27.9 ± 18.5	21.6 ± 11.2	23.3 ± 14.5	24.8 ± 16.0	0.8482	0.0508
AST (U/L)	23.0 ± 8.4	22.6 ± 9.7	24.6 ± 11.6	20.9 ± 7.1	21.9 ± 9.0	22.9 ± 10.0	0.5784	0.0605
SCr (umol/L)	67.1 ± 19.8	70.9 ± 20.7	68.4 ± 19.8	79.5 ± 20.2	70.4 ± 24.6	65.4 ± 22.7	0.2379	0.0506
BUN (umol/L)	5.86 ± 1.79	5.87 ± 3.38	5.69 ± 1.85	6.18 ± 3.15	6.32 ± 5.29	5.20 ± 2.00	0.9489	0.5616
UA (umol/L)	330.0 ± 82.3	326.4 ± 86.5	333.6 ± 85.6	329.9 ± 92.0	321.5 ± 81.1	309.7 ± 86.4	0.6094	0.1257

Data are presented as mean ± SD or median (interquartile range) values, *p* < 0.05. Q1, RBC lifespan of >75 days; Q2, RBC lifespan of 75–89 days; Q3, RBC lifespan of 90–111 days; and Q4, RBC lifespan of ≥112 days. ^a^
*p*: Healthy control vs. T2DM; ^b^
*p*: Q1 vs. Q2 vs. Q3 vs. Q4. Abbreviations: T2DM, type 2 diabetes mellitus; SBP, systolic blood pressure; DBP, diastolic blood pressure; Hb, hemoglobin; FBG, fasting blood glucose; P2BG, 2 h postprandial blood glucose; TG, triglyceride; TC, total cholesterol; LDL-c, low-density lipoprotein cholesterol; ALT, alanine aminotransferase; AST, aspartate transaminase; SCr, serum creatinine; BUN, blood urea nitrogen; UA, urine albumin.

**Table 2 jpm-12-01738-t002:** The correlation between the RBC lifespan and variables of T2DM.

Variables	*r*-Value	*p* Value
Hb (g/L)	0.08	0.089
Duration (years)	−0.077	0.108
HbA1c (%)	0.030	0.52
Age (years)	0.050	0.28
AG (mmol/L)	−0.33	<0.001
GA (%)	−0.14	0.01

Abbreviations: AG, average blood glucose; GA, glycosylated albumin; HbA1c, glycosylated hemoglobin.

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
