# Peer review of "Red Blood Cell Lifespan < 74 Days Can Clinically Reduce Hb1Ac Levels in Type 2 Diabetes"

_jpm, 2022, doi:10.3390/jpm12101738_

Round 1

Reviewer 1 Report

Dear Authors, 

Your MS is well written showing interesting evidences about an important aspect in the management of the most widespread endocrine disease, type 2 diabetes (T2DM). Given that HbA1c determines the clinical and therapeutic management of patients, changes in Hb concentration may alter this management. This represents the strength of the present study.

I have only minor suggestion. 

Introduction

I suggest adding references about the most important complication of diabetes:

-       Evaluation of Subclinical Vascular Disease in Diabetic Kidney Disease: A Tool for Personalization of Management of a High-Risk Population.

Kourtidou C, Rafailidis V, Varouktsi G, Kanakis E, Liakopoulos V, Vyzantiadis TA, Stangou M, Marinaki S, Tziomalos K.

J Pers Med. 2022 Jul 14;12(7):1139. doi: 10.3390/jpm12071139.

PMID: 35887636

-       Sex-specific effects of daily tadalafil on diabetic heart kinetics in RECOGITO, a randomized, double-blind, placebo-controlled trial.

Pofi R, Giannetta E, Feola T, Galea N, Barbagallo F, Campolo F, Badagliacca R, Barbano B, Ciolina F, Defeudis G, Filardi T, Sesti F, Minnetti M, Vizza CD, Pasqualetti P, Caboni P, Carbone I, Francone M, Catalano C, Pozzilli P, Lenzi A, Venneri MA, Gianfrilli D, Isidori AM.

Sci Transl Med. 2022 Jun 15;14(649):eabl8503. doi: 10.1126/scitranslmed.abl8503. Epub 2022 Jun 15.

PMID: 35704597

-       Effect of urinary albumin creatinine ratio on type 2 diabetic retinopathy and is cut-off value for early diabetic retinopathy diagnosis.

Wang X, Zhang M, Li T, Lou Q, Chen X.

Prim Care Diabetes. 2022 Aug 9:S1751-9918(22)00132-2. doi: 10.1016/j.pcd.2022.08.002. Online ahead of print.

PMID: 35961813

Reviewer 2 Report

Comments for the Author (Required):

The study by

 Zhou S et al, titled “Shorter red blood cell lifespan could cause clinical significance influence on HbA1c detection values in type 2 diabetes patients aims to understand the impact of RBC lifespan on HbA1c in type 2 diabetes patients in cervical cancer. They have shown the significant difference in FBG, TG, TC, LDLc levels and RBC lifespan in heathy control and T2DM patients. I have some concerns about this article.

Specific comments.

1.     Introduction: Page 3-4, the para runs like this “There are some reasons for this gap: for half a century, the traditional method of determining……………………………………………………of these methods. Page 4, A carbon monoxide…………………………………. RBC lifespan in patients”.

Author mentioned advantages of carbon monoxide (CO) breath test over other tests.

Since this method was reported others in Ref 7-10, it is not required in the introductions section.

2.     Methods:

2a. Page 5, the para runs like this”(2) received diagnosis and treatment at our hospital for at least 3 months, used a blood glucose monitoring system in the past 3 months, and underwent monitoring of finger blood glucose testing before and 2 hours after each meal, and before sleeping at least twice a week according to the A1C-derived Average Glucose (ADAG) study”.  This para is confusing to the readers need to be rewrite.

2b. Page 6, author mentioned “use of drugs that may affect the RBC lifespan, such as ribavirin, barbiturates, or phenobarbital sodium”- citation missing.

3.     Results:

3a. Page 10, table 1 legend; bP: Q1 vs. Q2 vs. Q3 vs. Q4- b should be superscript.

3b. Page 11, author need to cite the Figure 2A in the respected area.

3c. Page 11, the para runs like this “And as shown in Figure 2…..”- Figure 2 should be Figure 2B.

3d. Since author presented AG and HbA1c data in the Figure 3, the legend title needs to be changed

3e. Author claims that RBC lifespan greatly affect Hbc1 levels in T2DM patients, but in the Figure 3B, there is no significant difference among the group (Q1; Q2: Q3: Q4).

3f. Figure 4, author reported 14 % contribution of RBC lifespan contribution from Q1 group and 1 % contribution from Q2-Q4 groups.

The maximum lifespan of Q2 is 89 days and Q4 >112 days and the contribution are 0.9% and 1.13%. Although the difference in RBC lifespan between the groups is 23 days, the contribution is not significant.

3g. It would be nice to quantify AG and HbA1c in Heathy control vs T2DM and in Male vs Female T2DM patients

4.     Discussion:

Discussion need be to rewrite. Authors need to discuss previous findings with current finding, and conclusions in this section.

Reviewer 3 Report

The general Idea of this research is interesting. This manuscript describes the extent of the RBC lifespan of patients with T2DM affects the HbA1c detection value. Howerver,there are some questions still need to be solved:

1.   The variability of breath CO in an individual subject. The authors are certainly aware of this and thus performed all measurements early in the morning, which has been shown to be more consistent in previous studies. However, within day and week to week variability should be determined and discussed. If this was done in previous studies by the same authors, the variability seen using the current protocol should be discussed.

2.        The authors show statistically significant findings in a large group of subjects using an indirect measure of red cell lifespan and then suggest using the same method to make clinical decisions in individual diabetes patients. How accurate is this indirect measure of red cell lifespan? Is it recognized? This manuscript need to be illustrated and provided with appropriate references.

3.   Exclusion criterias:"(2) high blood glucose fluctuation; (3) repeated hypoglycemic attacks" - what does "high" in this case stand for? How many attacks? 

Round 2

Reviewer 2 Report

Comments for the Author (Required): 

Author need reformulate below para in the discussion,  

These results indicate that only shorter RBC lifespan could cause a clinically significant difference in HbA1c. These results indicate that only shorter RBC lifespan could cause a clinically significant difference in HbA1c. This phenomenon can be explained by the fact that the shorter RBC lifespan results in insufficient glycosylation of Hb exposed to certain blood glucose levels.

And this result also strongly indicated the HbA1c test values in T2DM with RBC lifespan less than 74d cannot truly reflect their blood glucose levels, resulting in chronic poor blood glucose control.

Page 17, para run like this …And this result also strongly indicated the HbA1c test values in T2DM with RBC lifespan less than 74d cannot truly reflect their blood glucose levels- It would be nice to maintain one format- either Days or d 

Author Response

Comments for the Author (Required): 

Author need reformulate below para in the discussion,  

These results indicate that only shorter RBC lifespan could cause a clinically significant difference in HbA1c. These results indicate that only shorter RBC lifespan could cause a clinically significant difference in HbA1c. This phenomenon can be explained by the fact that the shorter RBC lifespan results in insufficient glycosylation of Hb exposed to certain blood glucose levels.

And this result also strongly indicated the HbA1c test values in T2DM with RBC lifespan less than 74d cannot truly reflect their blood glucose levels, resulting in chronic poor blood glucose control.

Page 17, para run like this …And this result also strongly indicated the HbA1c test values in T2DM with RBC lifespan less than 74d cannot truly reflect their blood glucose levels- It would be nice to maintain one format- either Days or d 

ResponseThanks for the reviewer’s constructive suggestion, we have modified the content as:

  1. These results indicate that only a shorter RBC lifespan could cause a clinically significant difference in HbA1c, which can be explained by the fact of insufficient glycosylation of Hb exposed to certain blood glucose levels.

  1. And this result was consistent with our study, both of which strongly indicated the HbA1c test values in T2DM with a RBC lifespan less than 74 days cannot truly reflect their blood glucose levels, resulting in long-term poor blood glucose control.

  1. We have modified “74d” as “74 days” to maintain one format throughout our manuscript.

And this result was consistent with our study, both of which strongly indicated the HbA1c test values in T2DM with a RBC lifespan less than 74 days cannot truly reflect their blood glucose levels, resulting in long-term poor blood glucose control.